# Study on the Influence of Host–Guest Interaction on Tourists' Pro-Environment Behavior: Evidence from Taishan National Forest Park in China

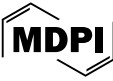

**Feifei Lu** [1], **Bingnan Wang** [1], **Juan Bi** [2],* and **Weiya Guo** [1],*

[1] College of Economics and Management, Qingdao Agricultural University, No. 700 Changcheng Road, Chengyang District, Qingdao 266109, China; feifei.lyu@qau.edu.cn (F.L.); bn_wang@stu.qau.edu.cn (B.W.)
[2] School of Tourism and Geography Science, Qingdao University, No. 308 Ningxia Road, Shinan District, Qingdao 266071, China
* Correspondence: bijuan@qdu.edu.cn (J.B.); guoweiya@qau.edu.cn (W.G.)

**Abstract:** This study explores the influence of host–guest interaction on tourists' pro-environment behavior. On the one hand, the experience attribute of host–guest interaction's influence on tourists' experiential value is sorted out. On the other hand, the relationship attribute of host–guest interaction's activation effect on tourists' personal norms is identified. Based on social exchange theory and normative activation theory, a structural equation model was established to depict the transmission mechanism from host–guest interaction to tourists' pro-environment behavior. The data were collected from tourists in Taishan National Forest Park (*n* = 499). The results indicated that host–guest interaction quality activated tourists' personal norms through consequence awareness, thus promoting tourists' pro-environment behavior. Meanwhile, the quality of host–guest interaction positively influenced tourists' pro-environment behavior through emotional experience value. Host–guest interaction quantity promoted tourists' pro-environment behavior by activating personal norms through responsibility ascription. In addition, although frequent host–guest interactions can enhance the social experience and functional experience value of tourists, the latter two cannot stimulate tourists' pro-environmental behavior. This study provides practical implications for promoting the sustainable development of national forest parks.

**Keywords:** national forest park; host–guest interaction; tourists' pro-environment behavior; sustainable development





## 1. Introduction

We are facing many environmental problems worldwide, such as air pollution, water pollution, global warming, etc. [1]. Especially with the rapid advancement of industrialization and urbanization, in some areas, the natural environment and resources are being degraded [2–4]. In this context, ecotourism has developed rapidly [5]. More and more people travel to natural places such as forest parks, wildlife parks, and the seacoast. Meanwhile, tourists' behavior also directly or indirectly affects the sustainable development of tourist destinations. Positive tourist behaviors, such as actively picking up garbage and respecting local customs and habits, not only help reduce the environmental pressure on the destination and promote ecological protection and cultural heritage but also help improve the tourist experience, thus promoting the sustainability of destination tourism. Negative tourist behavior, such as irresponsible destruction and pollution of the environment, may lead to environmental degradation and cultural alienation and may even threaten the long-term attractiveness of tourist destinations. Destinations use a variety of means to reduce the negative impacts of tourism, such as monitoring the volume of tourist traffic [6]. However, the maintenance of environmental protection in tourism cannot be separated from the extensive performance of pro-environmental behaviors by tourists [7]. Moreover,

pro-environmental behavior has been identified as an effective means to alleviate environmental problems [8]. Studying the driving factors of tourists' pro-environment behavior is valuable for destinations to formulate targeted strategies and measures to guide tourists' behavior, reduce carbon emissions, and promote the sustainable development of national forest parks.

Empirical research on pro-environmental behavior began in the 1970s and has continued to develop [9–11]. As more scholars join in, more attention has been paid to the interaction between human activities and the environment. The concept of "pro-environmental behavior" was introduced to describe behaviors that can reduce ecological harm, protect natural resources, and improve the natural environment [12,13]. Pro-environmental behavior is essentially considered a kind of altruism [14]. In addition, other concepts can also be used to describe prosocial behavior, such as "environmentally responsible behavior" [15], "ecological behavior" [16], and "environmentally supportive behavior" [17].

Tourists rarely have spontaneous pro-environmental behavior and willingness [18]. More and more studies focus on the driving factors of tourists' pro-environmental behavior [19], which can be roughly divided into tourist factors and destination factors. In terms of tourists' factors, Gifford et al. [20] found that childhood experience, knowledge and education, personality and self-interpretation, sense of control, values, politics, and world outlook, goals, perceived responsibility, cognitive bias, place attachment, age, gender, and choice activities would affect tourists' pro-environment behaviors and their concern for pro-environmental behaviors. Meyer [21] found that values and identity can stimulate tourists' pro-environmental behaviors [22]. Scholars also predict tourists' pro-environmental behavior at the destination based on tourists' daily behavior. For example, research has found that if tourists' pro-environment behavior at home is higher than the average level, they will also have better pro-environmental behavior at the tourist destination [23]. Xu et al. [24] found a significant positive correlation between pro-environmental behavior in tourists' origin places and pro-environmental behavior while traveling. Barr [25] argued that those who are the most environmentally friendly at home are still likely to use the most environmentally friendly means of transportation when traveling. In terms of tourist destination factors, Ling and Xu [26] believe that an individual's environment is an important factor in changing personal motivation for pro-environmental behavior. Lee et al. [23] found that when destination social responsibility increases during tourism, the impact of personal norms on pro-environmental behavior will weaken, highlighting the important role that destination environmental responsibility plays in influencing tourists' pro-environment behavior.

As mentioned above, although there have been studies on the driving factors that influence pro-environment behavior, there is currently a lack of research from the perspective of host–guest interaction. In fact, tourists interact with local residents during tourism, and this process will impact tourists' behavior. Previous research has found that the interaction between host and guest can have an impact on tourists' attitudes and behaviors and lead to positive or negative results [27,28]. For instance, residents' negative stereotypes about tourists could be reinforced during their encounters with tourists, leading to their hostility towards tourists [29]. Some scholars also believe that the impact of host–guest interaction on tourist behavior is related to the frequency and intensity of the interaction [30]. As an important part of tourism, host–guest interaction is likely to have an impact on tourists' pro-environmental behavior. However, its transmission mechanism remains to be explored.

The interaction between host and guest is a common and vital form of interaction in the process of national forest park tourism [31]. Many scholars have studied the interaction between host and guest. Pizam et al. [30] studied the degree of interaction between tourists and hosts and found that the deeper the interaction, the more positive emotions tourists have towards the host and the higher their satisfaction with their tourism experience. Sharpley [31] proposed that when hosts and tourists have direct cooperative relationships or frequent contact, a good tourism experience will often occur during the interaction process, affecting the attitudes, perceptions, and behaviors of hosts and tourists. Shi et al. [32]

discussed the benefits brought by the interaction between host and guest to local residents from the perspective of residents. The results found that host–guest interaction has a positive impact on various areas of young tourists' quality of life, including physical health and social relationships. However, existing research rarely pays attention to the impact of host–guest interaction on tourists' pro-environmental behavior. In tourist destinations, the interaction between tourists and local residents is vital to enriching their travel experience. This interaction not only allows tourists to have a deeper understanding of local culture and lifestyle but also enhances the perceived value of their experience. Based on social exchange theory, when tourists receive kindness and help from local residents, they may feel obligated to reciprocate. This reciprocal principle may not only be expressed as gratitude and respect for local residents but may also be translated into practical actions, such as participating in local public welfare activities or making pro-environmental choices. In this way, visitors can enjoy local culture and natural beauty while also contributing to sustainable local development. In addition, interactions with local residents may also inspire tourists' sense of moral responsibility to protect the environment. Based on the norm activation theory, by understanding local environmental issues and challenges, tourists can realize the importance of protecting the environment, stimulate awareness of the consequences of environmental issues and the attribution of responsibility, and then promote tourists' pro-environmental behavior through personal norms. Therefore, host–guest interaction may inspire tourists' pro-environmental behavior in two aspects: rational experience value perception and individual moral norm activation.

In summary, this study aims to explore the transmission mechanism from host–guest interaction to tourists' pro-environment behavior in the context of national forest park tourism. More specifically, based on social exchange theory, this paper first explores the impact of host–guest interaction on tourists' pro-environment behavior through experiential value. Second, based on normative activation theory, we explore the influence of host–guest interaction on the pro-environment behavior of tourists through the activation of personal norms. Third, from the perspective of host–guest interaction, this paper proposes specific practical suggestions for promoting tourists' pro-environment behaviors in national forest parks and their communities.

## 2. Literature Review and Hypothesis Development

### 2.1. Pro-Environment Behavior

Pro-environment behavior refers to the behavior of individuals to minimize the negative impact on the ecology or the behavior that is beneficial to the environment, with the intention of reducing the environmental burden through effective individual behavior [33]. Tourists' pro-environment behavior not only helps to improve their own quality of life but also reduces pollution and damage to the environment, helps promote the development of tourism in a sustainable direction, and achieves a coordinated symbiosis of economy, society, and environment. In existing research, human activities are considered to be an important factor in the rapid deterioration of the environment [34]. The damage level to components of the forest environment (such as soil and young trees) caused by tourist and recreational use was considered high [35]. Therefore, exploring the precursor factors of tourists' pro-environmental behavior is crucial to environmental protection in tourist destinations. Scholars have discussed it from different perspectives in the past. Hansmann [36] investigated the pro-environmental behavior of students, academic staff, technical staff, and administrative staff and found that those who have made progress in academic levels will have higher levels of pro-environmental behavior, but tourists who express pro-environmental intentions will not be converted into actual pro-environmental behavior [37]. Liu et al. [38] discovered that perceived environmental quality was an important antecedent of tourists' attitudinal factors towards tourists' pro-environmental behavioral intentions. Lavergne [39] studied the impact of perceptions of government approaches to environmental regulation on environmental motivation and the frequency of self-reported pro-environmental behaviors. Masud [40] studied whether attitudes toward

climate change, subjective norms, and perceived behavioral control are significantly associated with behavioral intentions to adapt to climate change and adopt pro-environmental behaviors. In contrast, host–guest interaction is rarely considered an antecedent factor in tourists' pro-environmental behavior. In fact, the impact of host–guest interaction on tourists' pro-environmental behavior cannot be ignored. On the one hand, interaction with local residents is an important part of tourists' travel experiences. By interacting with local residents, tourists will have the opportunity to understand and experience the life and culture of the local people. It helps to enhance the value of their tourism experience and stimulate tourists' pro-environmental behavior based on the reciprocal principle of social exchange theory. On the other hand, interaction with local residents, as a type of interpersonal relationship, may inspire tourists' personal norms about the environment from a moral level, thereby leading to behaviors that are conducive to environmental protection at the destination. To sum up, it is necessary for us to more comprehensively sort out the transmission mechanism of host–guest interaction on tourists' pro-environment behavior.

### 2.2. Relationship between Host–Guest Interaction and Tourists' Pro-Environment Behavior

The study of host–guest interaction began in the first half of the twentieth century. The interaction and impact between tourists and residents in tourism activities are complex. Scholars have conducted in-depth discussions on the quantity and quality of host–guest interactions in tourism activities and put forward many valuable opinions, forming a large number of research results. Wang et al. [40] first studied the impact of social interaction between hosts and guests on tourists' environmentally responsible behavior in a tourism context. The results found that tourists' environmental knowledge and environmental sensitivity are largely affected by the host's interactive behavior. In order to improve the surrounding living environment, the hosts will guide tourists' environmental awareness and encourage them to implement environmentally responsible behaviors. Tourists may implement behaviors that are beneficial to environmental protection due to their interactions with residents during the tourism process. For example, when the host and tourists have a direct cooperative relationship or have frequent contact, a good tourism experience will usually be generated during the interaction, thus affecting tourists' attitudes and behaviors [31]. However, the study of Wang et al. [41] aimed to explore how residents can actively promote tourists' pro-environment behavior. It adopted a qualitative research method and did not empirically testify to the relationship between host–guest interaction and tourists' pro-environmental behavior. Tu et al. [42] regard host–guest interaction as the exchange of social and emotional resources between host and guest. The core focus is on the emotional factor in host–guest interaction, namely gratitude, and does not explore the experiential attributes and moral activation functions of host–guest interaction. This article will establish a relationship model between host–guest interaction and tourists' pro-environmental behavior based on social exchange theory and the norm activation model.

#### 2.2.1. Social Exchange Theory

The American scholar Homans first proposed the social exchange theory. This theory regards social interaction behavior as a way of facilitating commodity exchange. The commodities here include both material commodities and non-material commodities. People compare the cost of social interaction with the benefit, form a value judgment, and, according to that, decide their subsequent behavior. Tourism is considered to be an activity in which tourists gain a positive experience. Only through physical or psychological experience can tourists perceive the value of tourism [43]. In this case, the perceived value is the perceived experience value. Wei et al. [44] found that experience value can be co-created by customers and others in interactions. Tourists realize the exchange of various resources in their interactions with residents, such as money, space, and emotions. This exchange enriches the tourists' experience value and affects the tourists' behavior. Therefore, it is reasonable to use social exchange theory to explore the impact of host–guest interaction on tourists' pro-environmental behavior.

Human beings are highly social, and individuals cannot live in isolation from society. While tourists enjoy the scenery during their journey, they inevitably interact with local residents. As an important part of tourism, host–guest interaction will affect tourists' experience value. According to Sánchez et al. [45], perceived experience value contains three dimensions, namely functional value, emotional value, and social value. Among them, functional value is one of the basic values of experience value [46], which refers to consumers' rational consideration of the quality of products and services [47] and is the main driving force for consumer decision-making [48]. Tourists pass their demand information to the host through interaction, and the host provides them with corresponding products and services to meet the tourists' needs, thereby obtaining the functional value of the tourism experience [49]. The higher the frequency of host–guest interaction, the better the quality, the more fully the information is conveyed, and the easier it is to enhance tourists' sense of functional value. Both the quantity and quality of host–guest interactions have an impact on tourists' perceived value. Emotional value refers to some emotions triggered by tourists in tourism activities, including physical or mental relaxation, happiness, etc. High-frequency and high-quality interaction can promote understanding between tourists and residents, deepen the emotional connection between the two sides, and enhance the emotional experience value of tourists. In this study, social value refers to the friendly attitude and enthusiastic service shown by local residents through tourism, which will satisfy tourists' needs in interpersonal relationships, social image, and social status and generate a kind of self-identity recognition and pride. Frequent and positive interactions with local residents can help residents gain a sense of self-efficacy and a sense of being respected and recognized in such interpersonal communication, thus enhancing their social value experience of tourism. Previous studies have also confirmed that host–guest interaction will enhance tourists' experience value (e.g., Wei et al., 2020 [44]); therefore, this article puts forward the following hypotheses:

**H1.** *Host–guest interaction has a positive impact on tourists' functional value.*

**H1a.** *Interaction quantity has a positive impact on tourists' functional value.*

**H1b.** *Interaction quality has a positive impact on tourists' functional value.*

**H2.** *Host–guest interaction has a positive impact on tourists' emotional value.*

**H2a.** *Interaction quantity has a positive impact on tourists' emotional value.*

**H2b.** *Interaction quality has a positive impact on tourists' emotional value.*

**H3.** *Host–guest interaction has a positive impact on tourists' social value.*

**H3a.** *Interaction quantity has a positive impact on tourists' social value.*

**H3b.** *Interaction quality has a positive impact on tourists' social value.*

Based on the reciprocal principle of social exchange theory, when a tourist interacts with local residents, they will expect to receive some kind of reward or gain. If this reward is positive, then the person will develop a positive attitude or behavior. On the contrary, if the reward is negative, such as being deceived or treated unfairly, then the person may no longer trust the tourist destination or even behave negatively. Therefore, when tourists' perceived experience value is higher, they are more inclined to engage in feedback behaviors towards the destination. For example, previous studies found that tourists' experience value promotes tourists' value co-creation behavior at the destination [44,50]. Therefore, we infer that tourists' experience value has a positive impact on their pro-environmental behavior.

**H4.** *Tourists' functional value has a positive impact on their pro-environmental behavior.*

**H5.** *Tourists' emotional value has a positive impact on their pro-environmental behavior.*

**H6.** *Tourists' social value has a positive impact on their pro-environmental behavior.*

2.2.2. Norm Activation Theory

Norm activation theory is a commonly used theory in studying pro-environmental behavior. The theory contains four elements, namely awareness of consequences, the ascription of responsibility, personal norms, and pro-environmental behavior. Consequence awareness refers to a person's awareness that not performing a certain behavior may have adverse consequences for others [51], that is, the individual's awareness of adverse consequences for the environment if he does not perform pro-environmental behavior. Responsibility ascription refers to tourists' sense of responsibility for the negative consequences of their actions on the environment. When a person realizes that their actions have had a negative impact on the environment, they tend to blame themselves for the consequences. Personal norms refer to the perception of an individual's moral responsibility for performing actions. Regarding the relationship between the four elements, according to norm activation theory, awareness of consequences and ascription of responsibility are two prerequisite factors that affect personal norms, and personal norms affect individual behavioral intention [52]. Consequence awareness, responsibility ascription, and personal norms are three important predictor variables in norm activation theory. When personal norms serve as mediating variables, responsibility ascription and consequence awareness serve as antecedent variables, acting on personal norms. Onwezen et al. [53] found that personal norms can directly or indirectly affect people's pro-environmental behavior. Joanes [54] found that consumers' perception and ascription of responsibility for the adverse consequences of clothing production and consumption will significantly and positively affect their personal norms, which in turn affects their willingness to reduce clothing consumption. Furthermore, Klöckner et al. [55] found that awareness of consequences has a direct positive impact on consumers' personal norms for choosing organic food. According to existing research, individuals will trigger personal norms because they are aware of the potentially harmful consequences of certain behaviors and have a sense of responsibility for them. When people's personal norms are activated, they are more likely to take action to protect the environment. Accordingly, this article puts forward the following hypotheses:

**H7.** *Tourists' awareness of the consequences of environmental problems in scenic spots positively affects their personal norms.*

**H8.** *Tourists' responsibility ascription for environmental problems in scenic spots positively affects their personal norms.*

**H9.** *Tourists' personal norms regarding environmental issues in scenic spots positively affect their pro-environmental behavior.*

In addition, some studies have found that awareness of consequences has a positive impact on responsibility adherence. For instance, Gao et al. [56] found that Chinese tourists' awareness of environmental consequences will affect their ascribed responsibility and moral obligation for environmental protection. The greater their sense of responsibility for not protecting the environment, the more guilty they will feel for not protecting the environment. In addition to directly affecting personal norms, Zhang et al. [57] indicated that consequence awareness also indirectly affects personal norms through responsibility ascription. In other words, when tourists realize that their actions will not protect or even damage the environment, they are more likely to attribute responsibility to themselves.

Accordingly, this article puts forward the following hypotheses:

**H10.** *Tourists' awareness of the consequences of environmental problems in national forest parks positively affects their responsibility ascription.*

People's sense of ethics, morality, and responsibility for the environment might be aroused during their interactions with other people [58]. Wu et al. [59] proposed that tourists develop an understanding and love for local people through interaction with local people, which can improve tourists' behavioral awareness and help them implement behaviors that maintain the environment. Wang et al. [41] found that hosts can adopt measures to interact with tourists frequently and positively, thus encouraging tourists to implement behaviors that protect the environment. For example, direct communication methods such as setting up signs with environmental protection content can improve tourists' willingness to protect the environment. Through frequent interactions, tourists have more opportunities to directly contact and understand local environmental issues, which provides tourists with a window to understand environmental issues. In addition, in the process of interaction with local residents, tourists establish a connection with the destination, are likely to link the environmental problems of the destination with themselves, and even regard themselves as members of the destination and ascribe responsibility for the environmental problems of the destination to themselves to a certain extent. Accordingly, this article puts forward the following hypotheses:

**H11.** *Interaction quantity has a positive impact on tourists' awareness of the consequences of environmental problems in scenic spots.*

**H12.** *Interaction quality has a positive impact on tourists' awareness of the consequences of environmental problems in scenic spots.*

**H13.** *Interaction quality has a positive impact on tourists' responsibility ascription for environmental issues in scenic spots.*

**H14.** *Interaction quantity has a positive impact on tourists' responsibility ascription for environmental problems in scenic spots.*

Based on the above analysis, this article proposes a theoretical model, as shown in Figure 1.

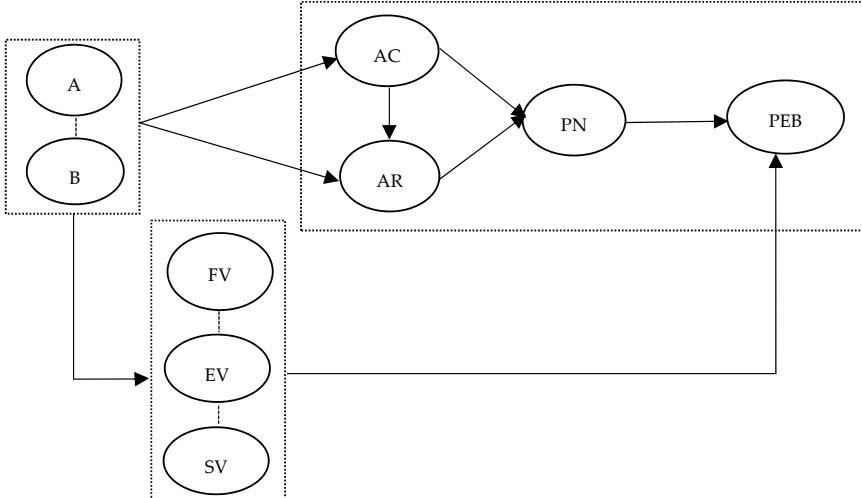

**Figure 1.** Conceptual model. Note: A = Host–guest interaction quantity; B = Host–guest interaction quality; AR = Ascription of responsibility; AC = awareness of consequences; PN = Personal norms; PEB = Tourists' pro-environment behavior; FV = Functional value; EV = Emotional value; SV = Social value.

## 3. Methods

### 3.1. Measurement of Constructs

This study uses a questionnaire survey to collect data. The questionnaire mainly consists of two parts. The first part is to measure the variables of the conceptual model, in which the measurement scales of each variable are derived from previous relevant literature and are measured by a five-level Likert scale. The items corresponding to the variables come from the previous relevant literature, are appropriately fine-tuned according to the tourism situation, and finally form a formal survey questionnaire. The quantity and quality of host–guest interaction were evaluated using seven items from Luo et al. [60,61]. The perceived experiential value of tourists was considered a three-dimensional concept: social value, emotional value, and functional value, which was assessed via twelve items from Sweeney et al. [62,63]. The scale of tourists' pro-environment behavior referred to the study of Su et al. [64], including six items. Tourists' awareness of environmental behavior can be divided into three dimensions: personal norms, awareness of consequences, and responsibility ascription. They were measured with eleven items from the scale of Han et al. [65]. The second part is the demographic information of the surveyed subjects, such as gender, age, education level, occupation, monthly income, permanent residence, marital status, and times of visits to Taishan National Forest Park.

### 3.2. Measurement Pretest

To ensure the validity of the content of the questionnaire, it was reviewed by two tourism scholars. Experts judge the relevance, clarity, and applicability of survey tools [64]. Pre-surveys are conducted prior to large-scale surveys. By collecting questionnaires on the Internet, 120 tourists who had been to Taishan National Forest Park were invited to fill out the questionnaires. The KMO value of the scale was >0.9, and the $p$-value was <0.05, indicating that the data were suitable for exploratory factor analysis. The results of exploratory factor analysis showed that except for the PEB2 factor load of the item's pro-environment behavior, which was 0.458, the factor load of each item was greater than 0.5, indicating an acceptable level of structural validity [66]. In addition, due to the acceptable Cronbach's alpha (all >0.7), every item should be retained except the pro-environmental behavior PEB2 item [67].

### 3.3. Data Collection and Sample Characteristics

The survey site of this study is Taishan National Forest Park in Tai'an City, Shandong Province, China, covering an area of 12,000 hectares. Taishan National Forest Park is a national scenic spot, a national AAAAA tourist attraction, and a World Heritage Site. Taishan National Forest Park is located in the east of the North China Plain, administered by Tai'an City, Shandong Province, with a total area of 11,868.6 hm$^2$. The main peak, Yuhuangding (117°6′ E, 36°15′ N), is 1532.7 m above sea level [68]. From January to October 2023, Taishan National Forest Park received a total of 8 million tourists, indicating an increase of 121.96% over the same period in 2019 in just four years. There are two reasons for choosing Taishan as the research site. One is that Taishan has a high reputation and can be used as a typical representative of the national forest park; the other is that the huge influx of tourists has brought great environmental pressure to Taishan National Forest Park. The research conclusion is of great significance for the sustainable development of tourism in Taishan National Forest Park. The research team distributed questionnaires to the tourists after climbing the mountain at the tourist exit of Taishan National Forest Park. A total of 600 questionnaires were distributed. Of these, 499 were completed and returned, representing a response rate of 83.3% [69]. The demographic information of the sample is shown in Table 1. There were more male respondents (60.3%) than female respondents (39.7%). In terms of the age of the respondents, 41.9% were between 21 and 30 years old, 23% were under 20, and 20% were between the ages of 31 and 40. About two-thirds of the respondents had a bachelor's or post-secondary education. In terms of occupation, students and corporate workers accounted for about half of the respondents.

Almost 28.9% of respondents have an income of 2000 yuan or less, and 25.6% have an income of 5001 to 8000 yuan. Among the respondents, 64.9% were unmarried, and most of them came to Taishan National Forest Park for the first time, with fewer respondents visiting the park three or more times. A total of 65.7% of the respondents came from cities outside of Shandong provinces, and 72.3% of them had junior college or undergraduate education degrees.

**Table 1.** Demographic description of samples (*n* = 499).

| Demographics | % | Demographics | % |
|---|---|---|---|
| Gender | | Income (RMB/Month) | |
| Male | 60.3 | 2000 and below | 28.9 |
| Female | 39.7 | 2001~5000 | 17.4 |
| Age | | 5001~8000 | 25.6 |
| 20 and below | 23 | 8001~10,000 | 12.6 |
| 21~30 | 41.9 | More than 10,001 | 15.5 |
| 31~40 | 20 | Residential region | |
| 41~50 | 8.2 | Other cities of Shandong province | 34.1 |
| 51 and over | 6.8 | Cities outside Shandong province | 65.7 |
| Education | | Foreign | 0.2 |
| Junior high school or below | 6.4 | Marital Status | |
| Senior high school/technical | 13.2 | Married | 32.9 |
| Junior college/undergraduate | 72.3 | Unmarried | 64.9 |
| Master's degree | 6.2 | Other | 2.2 |
| Master's degree above | 1.8 | Visiting frequency | |
| Occupation | | First time | 74.9 |
| Agricultural laborers | 1.6 | Second time | 13.6 |
| Enterprise staff | 26.6 | Third time | 4.2 |
| Government | 9.7 | Fourth time | 3.2 |
| Student | 32.3 | More than the Fourth time | 4.0 |
| Self-employed | 11.7 | | |
| Other | 18.1 | | |

## 4. Results

### 4.1. Common Method Variance Test

Common method variance (CMV) was tested through Harman's single factor test and confirmatory factor analysis (CFA) [70]. The Harman single-factor test results from exploratory factor analysis showed a multi-factor structure. The factor with the largest eigenvalue accounts for 21.313% (74.877%) of the total variance, which does not exceed half of the total variance explained. Therefore, there is no serious common method bias problem in this study [71].

### 4.2. Measurement Model Test

CFA was used to assess construct validity and estimate the model fit of the measurement model before testing the proposed hypotheses through structural equation modeling. The results show that the model has good fitting values. (X2/df = 3.132, RMSEA = 0.065, NFI = 0.880, IFI = 0.915, TLI = 0.903, CFI = 0.914). Reliability and validity were further tested by evaluating Cronbach's alpha, composite reliability (CR), convergent validity, and discriminant validity (Tables 2 and 3). The average variance extracted (AVE) of most latent variables was greater than 0.5, except for pro-environment behavior (0.459). However, if the AVE value is greater than 0.4, it will not have a serious impact on the test results. The combined reliability (CR value) is higher than the critical value of 0.6, indicating good convergent validity. According to Table 4, the value of the correlation coefficient between the two latent variables is smaller than the value of the square root of the latent variable AVE, indicating that the latent variables have good discriminant validity. The results show

that the measurement model is reliable and valid, allowing further hypothesis testing of the structural model.

**Table 2.** Confirmatory factor analysis of the measurement scale.

| Measurement Items | | Estimate | AVE | CR | α |
|---|---|---|---|---|---|
| **Host–guest interaction quantity** | | | | | |
| A1 | How often have you communicated with local residents? | 0.857 | | | |
| A2 | How often have you communicated with local service personnel in the tourism service scenario? | 0.713 | 0.595 | 0.814 | 0.812 |
| A3 | How often have you communicated with local residents outside of the tourism service scenario? | 0.736 | | | |
| **Host–guest interaction quality** | | | | | |
| B1 | My interaction with local residents is harmonious. | 0.914 | | | |
| B2 | My interaction with local residents is friendly. | 0.954 | 0.733 | 0.915 | 0.911 |
| B3 | My interaction with local residents is equal. | 0.846 | | | |
| B4 | My interaction with local residents is cooperative. | 0.686 | | | |
| **Social value** | | | | | |
| SV1 | Interaction with local residents makes me feel acceptable. | 0.911 | | | |
| SV2 | Interaction with local residents makes me feel that life is meaningful. | 0.876 | 0.797 | 0.922 | 0.921 |
| SV3 | Interaction with local residents makes me feel respected. | 0.89 | | | |
| **Emotional value** | | | | | |
| EV1 | Visiting Taishan National Forest Park gave me pleasure. | 0.851 | | | |
| EV2 | Visiting Taishan National Forest Park made me feel better. | 0.889 | 0.742 | 0.920 | 0.919 |
| EV3 | After visiting Taishan National Forest Park, my image of the Taishan National Forest Park was improved. | 0.867 | | | |
| EV4 | Taishan National Forest Park is a destination that I enjoy. | 0.837 | | | |
| **Functional value** | | | | | |
| FV1 | Interaction with local residents helped me better plan my itinerary. | 0.782 | | | |
| FV2 | I think Taishan National Forest Park is worth visiting. | 0.537 | | | |
| FV3 | Interaction with local residents helped me find the attractions I wanted to visit. | 0.875 | 0.588 | 0.874 | 0.868 |
| FV4 | Through interaction with local residents, I received products and services that better met my needs. | 0.878 | | | |
| FV5 | Interaction with local residents helped me solve some problems I encountered during my journey. | 0.708 | | | |
| **Awareness of consequences** | | | | | |
| AC1 | Irresponsible environmental behavior will lead to ecological degradation and natural resource consumption in Taishan National Forest Park Scenic Area. | 0.887 | | | |
| AC2 | Environmentally irresponsible behavior can possibly generate a huge environmental impact on nearby residents. | 0.83 | 0.715 | 0.909 | 0.893 |
| AC3 | Irresponsible environmental behavior can cause environmental deterioration in Taishan National Forest Park. | 0.914 | | | |
| AC4 | Responsible environmental behavior helps reduce negative impacts on the environment. | 0.742 | | | |
| **Ascribed responsibility** | | | | | |
| AR1 | I believe that every traveler is partly responsible for the Taishan National Forest Park's environmental problems. | 0.863 | | | |
| AR2 | I think every tourist in Taishan National Forest Park Scenic Area is partly responsible for the environmental problems caused by irresponsible environmental behavior in Taishan National Forest Park Scenic Area. | 0.798 | 0.643 | 0.843 | 0.845 |
| AR3 | I feel that every traveler is jointly responsible for the environmental deterioration caused by the development of tourism for Taishan National Forest Park. | 0.74 | | | |

**Table 2.** *Cont.*

| Measurement Items | | Estimate | AVE | CR | α |
|---|---|---|---|---|---|
| Personal norm | | | | | |
| PN1 | I feel that I have a moral responsibility to reduce the harm to the environment. | 0.806 | 0.733 | 0.916 | 0.912 |
| PN2 | No matter what others do, I will adhere to my values and principles and travel in a responsible way for the environment. | 0.895 | | | |
| PN3 | I think I should do things that are beneficial to the environment when traveling. | 0.845 | | | |
| PN4 | I think people should reduce the negative impact on the local community during travel. | 0.876 | | | |
| Pro-environment behavior | | | | | |
| PEB1 | I comply with the legal ways not to destroy the destination's environment. | 0.696 | 0.459 | 0.806 | 0.804 |
| PEB3 | When I see garbage, I will make an effort to put them in the trash can. | 0.575 | | | |
| PEB4 | If there are cleaning environment activities, l am willing to attend. | 0.537 | | | |
| PEB5 | I would convince my travel companions, if any, to protect the natural environment of the destination. | 0.742 | | | |
| PEB6 | I will not destroy the animals and plants in Taishan National Forest Park Scenic Area when visiting. | 0.802 | | | |

**Table 3.** Discriminant validity for the measurement model.

| Variables | Host–Guest Interaction Quantity | Host–Guest Interaction Quality | Social Value | Emotional Value | Functional Value | Awareness of Consequences | Ascribed Responsibility | Personal Norm | Pro-Environment Behavior |
|---|---|---|---|---|---|---|---|---|---|
| Host–guest interaction quantity | 0.771 | | | | | | | | |
| Host–guest interaction quality | 0.376 ** | 0.856 | | | | | | | |
| Social value | 0.337 ** | 0.521 ** | 0.892 | | | | | | |
| Emotional value | 0.180 ** | 0.350 ** | 0.506 ** | 0.861 | | | | | |
| Functional value | 0.334 ** | 0.422 ** | 0.592 ** | 0.531 ** | 0.766 | | | | |
| Awareness of consequences | 0.020 | 0.170 ** | 0.241 ** | 0.408 ** | 0.238 ** | 0.846 | | | |
| Ascribed responsibility | 0.120 ** | 0.188 ** | 0.273 ** | 0.415 ** | 0.319 ** | 0.634 ** | 0.802 | | |
| Personal norm | 0.041 | 0.224 ** | 0.262 ** | 0.450 ** | 0.281 ** | 0.777 ** | 0.582 ** | 0.856 | |
| Pro-environment behavior | 0.141 ** | 0.290 ** | 0.312 ** | 0.433 ** | 0.333 ** | 0.614 ** | 0.527 ** | 0.632 ** | 0.678 |

Note: ** $p < 0.01$. The items on the diagonal represent the square roots of the AVE, and off-diagonal elements are the correlation estimates.

**Table 4.** Hypothesis test results.

| Hypothesis | Path Relationships | | | Standardized Coefficient | SE | CR | *p* | Test Result |
|---|---|---|---|---|---|---|---|---|
| H1a | FV | <--- | A | 0.249 | 0.052 | 4.479 | *** | Supported |
| H1b | FV | <--- | B | 0.315 | 0.065 | 5.813 | *** | Supported |
| H2a | EV | <--- | A | 0.058 | 0.048 | 1.057 | 0.291 | Not supported |
| H2b | EV | <--- | B | 0.345 | 0.062 | 6.245 | *** | Supported |
| H3a | SV | <--- | A | 0.169 | 0.052 | 3.374 | *** | Supported |
| H3b | SV | <--- | B | 0.487 | 0.07 | 9.203 | *** | Supported |
| H4 | PEB | <--- | FV | 0.03 | 0.032 | 0.615 | 0.538 | Not supported |
| H5 | PEB | <--- | EV | 0.089 | 0.031 | 2 | * | Supported |
| H6 | PEB | <--- | SV | 0.033 | 0.029 | 0.665 | 0.506 | Not supported |
| H7 | PN | <--- | AC | 0.726 | 0.057 | 13.449 | *** | Supported |
| H8 | PN | <--- | AR | 0.216 | 0.042 | 4.58 | *** | Supported |
| H9 | PEB | <--- | PN | 0.847 | 0.051 | 13.871 | *** | Supported |
| H10 | AR | <--- | AC | 0.685 | 0.06 | 13.517 | *** | Supported |
| H11 | AC | <--- | A | −0.09 | 0.039 | −1.562 | 0.118 | Not supported |
| H12 | AC | <--- | B | 0.228 | 0.048 | 4.111 | *** | Supported |
| H13 | AR | <--- | B | 0.064 | 0.047 | 1.395 | 0.163 | Not supported |
| H14 | AR | <--- | A | 0.112 | 0.038 | 2.343 | * | Supported |

Note: * $p < 0.05$, *** $p < 0.001$.

### 4.3. Structural Model Test

Structural equation modeling (SEM) was utilized to test the hypothesized relationships. Table 4 shows the SEM results, and Figure 2 more intuitively shows the path relationships between the hypothetical variables. The results show that the structural model fits the data well ($X2/df$ = 2.941, RMSEA = 0.062, CFI = 0.920, IFI = 0.921, TLI = 0.911). The results supported nine direct relationships out of 14 hypotheses (Table 4).

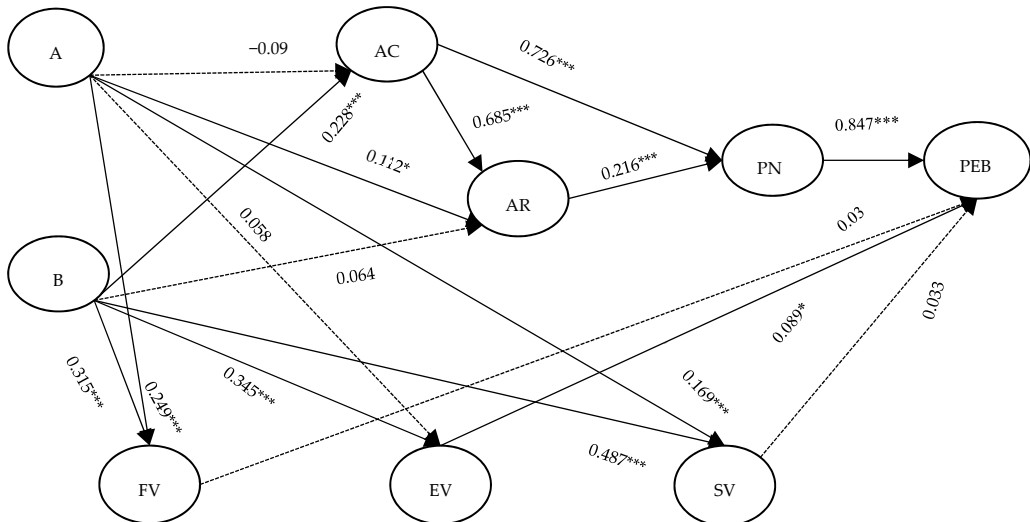

**Figure 2.** Structural equation modeling. Note: * $p < 0.05$, *** $p < 0.001$. A = Host–guest interaction quantity; B = Host–guest interaction quality; AR = Ascription of responsibility; AC = Awareness of consequences; PN = Personal norms; PEB = Tourists' pro-environment behavior; FV = Functional value; EV = Emotional value; SV = Social value.

H1a and H3a were supported. Host–guest interaction quantity has significantly positive impacts on tourist functional value ($\beta$ = 0.249, $p < 0.05$) and social value ($\beta$ = 0.169, $p < 0.05$). H2a was not supported. Host–guest interaction quantity was not significantly associated with tourist emotional value ($\beta$ = 0.058, $p > 0.05$). H1b, H2b, and H3b were supported. Host–guest interaction quality exerts a significantly positive influence on tourist functional value ($\beta$ = 0.315, $p < 0.05$), emotional value ($\beta$ = 0.345, $p < 0.05$), and social value ($\beta$ = 0.169, $p < 0.05$). H4 and H6 were not supported. Tourist functional and social values were not significantly associated with tourists' pro-environment behavior ($\beta$ = 0.033, $p > 0.05$; $\beta$ = 0.033, $p > 0.05$). Tourist emotional value was significantly and positively related to tourists' pro-environment behavior ($\beta$ = 0.089, $p < 0.05$). Thus, H5 was supported.

H7, H8, H9, and H10 were supported. Tourists' awareness of environmental problems has a significant positive impact on tourists' personal norms ($\beta$ = 0.726, $p < 0.05$). Tourists' responsibility ascription for environmental issues has a significant positive impact on tourists' personal norms for environmental issues ($\beta$ = 0.216, $p < 0.05$). Tourists' personal norms have a significant positive impact on their pro-environmental behavior ($\beta$ = 0.847, $p < 0.05$). Tourists' consequences awareness has a significant positive impact on their responsibility ascription for environmental problems ($\beta$ = 0.685, $p < 0.05$).

H11 was not supported. Host–guest interaction quantity does not significantly influence tourists' consequence awareness of environmental problems ($\beta$ = −0.09, $p > 0.05$). H14 was supported. Host–guest interaction quantity had a significant positive impact on tourists' responsibility ascription for environmental problems ($\beta$ = 0.112, $p < 0.05$). H11 was supported, while H13 was not supported. Host–guest interaction quality has a significant positive impact on tourists' awareness of environmental problems ($\beta$ = 0.228, $p < 0.05$). However, it does not have a significant impact on tourists' responsibility ascription for environmental problems ($\beta$ = 0.064, $p > 0.05$).

## 5. Conclusions, Discussion, and Implications

### 5.1. Conclusions

Based on social exchange theory and normative activation theory, we did a survey of tourists visiting Taishan National Forest Park to explore the influence of the host–guest interaction on tourists' pro-environment behavior. The results show that host–guest interaction quantity and quality have significant positive effects on both the functional value and social value of tourists. However, only interaction quality has a significant positive impact on tourists' emotional value. And it is only emotional value that could exert a significant positive impact on tourists' pro-environmental behavior. That is, the quality of host–guest interaction can promote tourists' pro-environment behavior through emotional experience value. This may be because functional value and social value are not enough to stimulate tourists' reciprocal psychology, and emotional value can induce gratitude and corresponding pro-environmental behaviors. In terms of the activation of moral sense, the quantity of host–guest interactions can significantly affect the ascribed responsibility of tourists, thus activating personal norms and promoting tourists' pro-environment behaviors; on the other hand, the quality of interactions can significantly affect tourists' awareness of consequences, activate personal norms, and promote tourists' pro-environment behaviors. The reason why the quality of interaction does not significantly affect the responsibility of tourists may be that individuals tend to shirk their responsibilities and rely on each other in friendly and intimate interactions. On the contrary, the number of interactions can significantly enhance the responsibility of tourists, activate personal norms, and promote tourists' pro-environmental behavior. The number of interactions did not significantly affect tourists' awareness of consequences, possibly because superficial interactions, even in large numbers, were not sufficient to help tourists gain a clear understanding of local environmental problems or the possible impact of individual actions on the environment. Above all, both the quantity and quality of host–guest interaction can promote tourists' pro-environment behavior through different paths.

### 5.2. Theoretical Contributions

There are three theoretical contributions in this paper. First of all, it clarifies the influence of host–guest interaction on tourists' pro-environment behavior. Starting from the experience attribute and relationship attribute of the interaction. It sorts out and tests its enhancement of tourists' experience value and the activation effect of personal norms. Previous studies have explored the impact of the interaction on tourists' attitudes or behaviors from the perspective of the experiential or relational attributes of the interaction, respectively [43,44]. There is a lack of systematic analysis of the mechanism of host–guest interaction on tourists' pro-environment behavior. Second, this paper enriches the study of host–guest interaction by exploring the quantitative and qualitative effects of subjective interaction. Previous studies focused on the quality of host–guest interaction and believed that positive host–guest interaction could have an impact on the behavior of tourists or residents (e.g., Xiong et al., 2021 [72]), but ignored the quantity of host–guest interaction. The results of this study show that the quantity of host–guest interactions itself has a significant positive impact on tourists' functional experience and social experience value and can effectively affect tourists' sense of belonging to responsibility, thus activating tourists' personal norms and, finally, promoting tourists' pro-environment behavior. Therefore, this study expands the research on the connotation and influence of the interaction between residents and tourists. Third, this study expands the normative activation theory model and introduces new antecedents, that is, host–guest interaction. Previous studies have encouraged the expansion of normative activation theory. Specifically, this study explores the influence of tourist–resident interaction on tourists' responsibility ascription and awareness of consequences, which offers a powerful supplement to the existing literature.

### 5.3. Practical Implications

The research conclusion of this paper has practical guiding significance for promoting the sustainable development of national forest parks. First of all, this study found that host–guest interaction can promote tourists' pro-environment behavior. Therefore, destination authorities should increase the opportunities for tourists to interact with local residents in various ways. For example, encourage local residents to participate in tourism operations or services, organize some free public welfare activities, and invite local residents to participate as volunteers. For example, some museums organize local residents as volunteer docents, and visitors can interact with these local residents while visiting. At the same time, the management department should also pay attention to the supervision of the quality of the interaction between the host and the guest and promote positive interaction between the host and the guest. Secondly, this study finds that emotional experience can enhance tourists' pro-environment behaviors. Therefore, it is necessary to enhance the value of tourists' emotional experiences. For example, destination residents take the initiative to provide tourists with information to help tourists better explore the destination, which helps to enhance tourists' sense of pleasure and thus stimulate more positive behaviors in tourists.

### 5.4. Limitations and Future Research Directions

This study also has certain limitations. Specifically speaking, firstly, this study takes Mount Tai as the research site, which only represents the type of destination, such as national forest parks. A single destination type will affect the generalization of the research conclusions. Second, this study only discusses the impact of host–guest interaction on tourists' pro-environment behavior from dimensions of the quantity and quality of host–guest interaction, but the connotation of host–guest interaction also includes other dimensions, such as interaction type and interaction intensity, which need to be further explored. Therefore, future studies can validate this model for other destination types and use qualitative methods to explore other dimensions of host–guest interaction. In addition, in the future, the interaction between host and guest can be included in the study of other attitudes and behaviors of tourists, such as tourist citizenship behavior.

**Author Contributions:** F.L.: conceptualization, design, and manuscript preparation. J.B. and W.G.: conceptualization, design, manuscript preparation, and supervision of this project. B.W. and F.L.: data preparation and design, formal analysis, original draft preparation, and manuscript preparation sections. Each author contributed to the conceptualization and writing of this paper. All authors have read and agreed to the published version of the manuscript.

**Funding:** This research was funded by the Forestry Science and Technology Development Project of China (Grant No. KJZXRZ202306), the Natural Science Foundation of Shandong Province (Grant No. ZR202103040840), the University Innovation and Technology Project of Shandong Province in China (Grant No. 2020RWG004), the High-level Research Foundation of QAU (Grant No. 1119710), and the Project Supported by Enterprises and Institutions (Grant No. 6602424706).

**Institutional Review Board Statement:** This study does not involve any ethical issues.

**Data Availability Statement:** The data from this research are publicly available.

**Conflicts of Interest:** The authors declare no conflicts of interest.

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
