# Peer review of "Study on the Influence of Host–Guest Interaction on Tourists’ Pro-Environment Behavior: Evidence from Taishan National Forest Park in China"

_forests, doi:10.3390/f15050813_

Round 1

Reviewer 1 Report

Comments and Suggestions for Authors

I found your article very interesting and important. Before accepting for publishing, I suggest a few modifications:

1. Is there any particular place (e.g. in China) that your would like to point out when stating: "...human living conditions are deteriorating in decades" (line 35-37).

2. Literature review section should be backed by specific examples and explained better. For instance, line 149-150, what did Liu et. al concluded in his/their research?

3. Line 173 and 183 mention the same reference (Wang), check if it is Wang or Wang et. al when referring to this article.

4. Line 205; "and other resource exchanges" might be excess here. Please review. The same in line 212 with "it concluded that". And line 331, "for details".

5. Check weird font used in lines 363-366.

6. Line 394; "...its environmental problems are particularly severe" - which ones in particular?

7. Line 408-409; The last sentence of the 3.3 section does not match other statements made in this section, e.g. about where the respondents come from and how many of them had a college or undergraduate education degree.

8. When you talk about the practical implications, you say that locals could participate more in tourism operations and services. Would you say that these kind of suggestions would be accepted by them easily?

Comments on the Quality of English Language

Minor editing of English is suggested. Be consistent in vocabulary (e.g. environmental vs. environment behaviour).

Author Response

Thanks so much for your helpful advice and comments. Please see below the detailed answers.

I found your article very interesting and important. Before accepting for publishing, I suggest a few modifications:

  1. 1. Is there any particular place (e.g. in China) that your would like to point out when stating: "...human living conditions are deteriorating in decades" (line 35-37).

Since it is a universal phenomenon that has been mentioned in previous researches, for instance, references[2-4], we are not meant to refer to a particular place.

  1. Literature review section should be backed by specific examples and explained better. For instance, line 149-150, what did Liu et. al concluded in his/their research?

The reference has added, please see the details below.

“Liu et al. [36] discovered that perceived environment quality was an important antecedent of tourists' attitudinal factors towards tourists’ pro-environmental behaviors. ”

  1. Line 173 and 183 mention the same reference (Wang), check if it is Wang or Wang et. al when referring to this article.

It  was modified, please see the text. “Wang et al. [39] first studied the impact of social interaction between hosts and guests on tourists' environmentally responsible behavior in a tourism context. ”

  1. Line 205; "and other resource exchanges" might be excess here. Please review. The same in line 212 with "it concluded that". And line 331, "for details".

As you suggested, we have deleted “and other resource exchanges”, “it concluded that”, and “for details”.

  1. Check weird font used in lines 363-366.

According to your advice, we have modified the font.

  1. Line 394; "...its environmental problems are particularly severe" - which ones in particular?

It was modified and displayed as follows.

“the other is that the huge influx of tourists has brought great environmental pressure to Taishan National Forest Park.”

  1. Line 408-409; The last sentence of the 3.3 section does not match other statements made in this section, e.g. about where the respondents come from and how many of them had a college or undergraduate education degree.

According to your advice, we added the explanations. And it was displayed as follows.

“65.7 percent of the respondents came from cities out of Shandong provinces, and 72.3 percent of them had junior college or undergraduate degree of education.”

  1. When you talk about the practical implications, you say that locals could participate more in tourism operations and services. Would you say that these kind of suggestions would be accepted by them easily?

It is hard to say. For some local residents, participating in tourism operations offers them a working opportunity. They would like to accept such suggestions. For some other residents, they may hesitate to participant in because of unfamiliarity. Therefore, we propose that destination authorities should encourage residents to participate in tourism operations and services.

Reviewer 2 Report

Comments and Suggestions for Authors

The article under consideration is devoted to the interesting topic of host-guest interaction in a selected Chinese forest park while focusing on the pro-environmental behavior of visitors. I consider the topic of visitor behavior concerning the host environment to be very important for many destinations in the world that are struggling with the negative impacts of tourism, so any quality analysis is welcomed by the professional community.

The article has a very good formal level: the literature search uses a reasonable amount of professional sources, the methodology is correctly designed, and the results are clearly presented and discussed. However, after reading the article, I have a slight feeling that statistical methods were too robust and the resulting conclusions too abstract for successful use in practice. However, I do not want to diminish the scientific value of the article, which is good.

My minor reservations are directed toward the presentation of the results of statistical analyses when, due to the frequent use of abbreviations, I lost track of key concepts and statistical indicators. I flipped through the article too often to "decode," e.g., Figure 2, tables, etc. For the results of statistical analyses, I would then expect standard abbreviations to be used, such as small (not capital) p for probability, etc. I think that small changes in this respect would significantly contribute to the clarity of the results.

Author Response

Many thanks  for your advice and kind comments. Please see below the detailed answers.

In table 2, we added a line above each variable measurement item to list the full name of the variables.

In table 3, we presented the full name of the variables.

We added explanation of the abbreviations in the note part under the figure.

We changed capital p to small found in the table. Please see in the text.

Reviewer 3 Report

Comments and Suggestions for Authors

The introduction is described in great detail. Host-guest interactions in tourism have been well characterized, supported by available literature.

Lines 46-47, it is worth adding here that exceeding the recreational potential of forests, especially in national forest parks, will increase the risk of damage to components of the forest environment. Therefore, the volume of tourist traffic in national parks must be monitored and limited under certain conditions (see: https://doi.org/10.1007/s11629-016-4018-z).

The aim of the study was clearly formulated in 3 points. Later in the study, 14 main research hypotheses were formulated. The proposed model makes it easier to understand the experiment as a whole, which is not easy considering the 14 research hypotheses being tested. Research planned in this way proves that the entire experiment has been carefully thought through, making it easier to control and draw the right conclusions.

Line 141-142, it is worth adding that this also applies to forests made available via tourist trails: The damage level of the examined components of the forest environment (soil, young trees) determined in the study as a result of tourist and recreational use should be considered as high (https://doi.org/10.26202/sylwan.2019101).

To achieve the complex goal of the study, a survey was constructed based on available literature sources, using rating scales for individual questions commonly used in this type of research.

What is surprising, however, is the large percentage of surveys that were not successful and were not included in the study (17%). The reason may be the question about income, but why this question if the results do not take into account the importance of income on tourists' behavior?

Tables 2 and 3: although the symbols used in the table are explained in the text or given in the description of Fig. 1, I recommend that in the description of the tables the abbreviations used in them be expanded, e.g.: AVE, CR, SV, EV...

The conclusion answers the questions included in the purpose of the study and highlights the most important results obtained. This is a model layout of a scientific article.

The practical implications of the results obtained and the limitations of this research were very well defined.

To sum up, I believe that the study presents original research results, important from the point of view of the growing demand for tourism and recreation in forests and national parks. The obtained results of host-guest interactions are particularly important for national parks, especially newly established ones. It is necessary to convince the hosts that it is worth protecting a given area, and that sustainable tourism can bring many benefits to the inhabitants of this area. In turn, tourists receive better conditions for their stay in a place with a special nature and landscape. The results provide important information for national park managers who can implement favorable regulations that promote appropriate host-guest interactions.

In my opinion, the study may be of interest to a wider group of readers.

Author Response

The introduction is described in great detail. Host-guest interactions in tourism have been well characterized, supported by available literature.

  1. Lines 46-47, it is worth adding here that exceeding the recreational potential of forests, especially in national forest parks, will increase the risk of damage to components of the forest environment. Therefore, the volume of tourist traffic in national parks must be monitored and limited under certain conditions (see: https://doi.org/10.1007/s11629-016-4018-z).

The aim of the study was clearly formulated in 3 points. Later in the study, 14 main research hypotheses were formulated. The proposed model makes it easier to understand the experiment as a whole, which is not easy considering the 14 research hypotheses being tested. Research planned in this way proves that the entire experiment has been carefully thought through, making it easier to control and draw the right conclusions.

Thanks for your advice. We understand that monitoring the volume of tourist traffic is an important method in controlling the risk of environment damage. However, we want to emphasize the significance of stimulating tourists’ pro-environment behavior in this paper. Therefore, we added this reference in the manuscript as an instance in explaining how to reduce negative impacts of tourism in previous studies. It was displayed as follows.

Destinations use a variety of means to reduce the negative impacts of tourism, such as monitoring the volume of tourist traffic ( Dudek, T. (2017).

  1. Line 141-142, it is worth adding that this also applies to forests made available via tourist trails: The damage level of the examined components of the forest environment (soil, young trees) determined in the study as a result of tourist and recreational use should be considered as high (https://doi.org/10.26202/sylwan.2019101).

To achieve the complex goal of the study, a survey was constructed based on available literature sources, using rating scales for individual questions commonly used in this type of research.

As you suggested, we added the reference to further describe the sever damage to the environment that was caused by tourist behaviors. It was displayed as follows.

The damage level to components of the forest environment (such as soil, young trees) caused by tourist and recreational use was considered high (Dudek, et al., 2020)

  1. What is surprising, however, is the large percentage of surveys that were not successful and were not included in the study (17%). The reason may be the question about income, but why this question if the results do not take into account the importance of income on tourists' behavior?

This is a good question. There were mainly two reasons that 17% of the questionnaires was deleted. First, some respondents left out some questions. Second, some respondents gave the same score to all the questions. However, 70% and above is an acceptable response rate for survey study (Shannon,1948).  

As for the reason about income question, income is generally considered as a part of demographic information about the respondents. It provides knowledge about the tourists visiting Taishan National Forest Park, just like education level, sex and age.

  1. Tables 2 and 3: although the symbols used in the table are explained in the text or given in the description of Fig. 1, I recommend that in the description of the tables the abbreviations used in them be expanded, e.g.: AVE, CR, SV, EV...

In table 2, we added a line above each variable measurement item to list the full name of the variables.

In table 3, we presented the full name of the variables.

We added explanation of the abbreviations in the note part under the figure. 

The conclusion answers the questions included in the purpose of the study and highlights the most important results obtained. This is a model layout of a scientific article.

The practical implications of the results obtained and the limitations of this research were very well defined.

To sum up, I believe that the study presents original research results, important from the point of view of the growing demand for tourism and recreation in forests and national parks. The obtained results of host-guest interactions are particularly important for national parks, especially newly established ones. It is necessary to convince the hosts that it is worth protecting a given area, and that sustainable tourism can bring many benefits to the inhabitants of this area. In turn, tourists receive better conditions for their stay in a place with a special nature and landscape. The results provide important information for national park managers who can implement favorable regulations that promote appropriate host-guest interactions.

In my opinion, the study may be of interest to a wider group of readers.

Thanks again for your kind comments.

Reviewer 4 Report

Comments and Suggestions for Authors

Forests -2931967Study on the Influence of Host-Guest Interaction on Tourists’ Pro-Environment Behavior Evidence from Taishan National Forest Park in China

This is a well written concise research article, but I have a few suggestions for improvement of the manuscript.

Introduction

Lines 88-89- would be good to give examples of possible and negative results.

Line 128-want is “experimental value”? may want to rephrase.

Literature review & Hypothesis development – nicely done.

Methods

Lines 357-369 -authors could put a chart in the Appendix for all parameters mentioned in Measurement of Constraints

Lines 386-388 – need location figure for Taishan National Forest Park

Results

Line 485- Figure 2 legend- need to spell out A, B, FV, AC, AR, EV, PN, SV, FEB

Conclusions, Discussion and Implications

I would split into two sections- Discussion (lines 490-538) and the rest in Conclusions.

Comments on the Quality of English Language

The English usage needs some minor editorial corrections.

Author Response

This is a well written concise research article, but I have a few suggestions for improvement of the manuscript.

Thanks  for your advice and kind comments. Please see below the detailed answers.

Introduction

  1. Lines 88-89- would be good to give examples of possible and negative results.

We added an example there. For instance, residents' negative stereotypes about tourists could been reinforced during their encounter with tourists, leading to their hostility towards tourists (Chen et al., 2018).

  1. 2. Line 128-want is “experimental value”? may want to rephrase.

We checked and found it was experiential value in the manuscript.

Literature review & Hypothesis development – nicely done.

Methods

  1. 3. Lines 357-369 -authors could put a chart in the Appendix for all parameters mentioned in Measurement of Constraints

In the paper, the issues related to demographic information were listed in table 1. And table 2 presented the measurement items of each variable. So t seems that there is no need to put a separate form in the Appendix.

  1. 4. Lines 386-388 – need location figure for Taishan National Forest Park

Since we can't get the rights to the images,we added the explanations for the location of  Taishan National Forest Park.  And it was displayed as follows.

Taishan National Forest Park is located in the east of the North China Plain, administered by Tai'an City, Shandong province, with a total area of 11868.6 hm². The main peak, Yuhuangding (117°6 'E, 36°15' N), is 1,532.7 meter above sea level[68].

Results

Line 485- Figure 2 legend- need to spell out A, B, FV, AC, AR, EV, PN, SV, FEB

We have added the explanations under Fig.2, please see in the text.

Conclusions, Discussion and Implications

  1. I would split into two sections- Discussion (lines 490-538) and the rest in Conclusions.

Thanks for your advice. However, we believe that this section  is logically one whole, so we did not split into two sections.
